# Centre vortex structure of QCD-vacuum fields and confinement

**Derek Leinweber⋆, James Biddle and Waseem Kamleh**

Centre for the Subatomic Structure of Matter, Department of Physics,
University of Adelaide, SA 5005, Australia

⋆ derek.leinweber@adelaide.edu.au

*Proceedings for the XXXIII International Workshop on High Energy Physics,
Hard Problems of Hadron Physics: Non-Perturbative QCD & Related Quests
ONLINE, 8-12 November 2021*

## Abstract

The non-trivial ground-state vacuum fields of QCD form the foundation of matter. Using modern visualisation techniques, this presentation examines the microscopic structure of these fields. Of particular interest are the centre vortices identified within the ground-state fields of lattice QCD. Our current focus is on understanding the manner in which dynamical fermions in the QCD vacuum alter the centre-vortex structure. The impact of dynamical fermions is significant and provides new insights into the role of centre vortices in underpinning both confinement and dynamical chiral symmetry breaking in QCD.

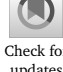

## 1 Introduction

Lattice QCD calculations have been instrumental in revealing the fundamental role of centre vortices [1–12] in the ground-state vacuum fields in governing the confinement of quarks.

By identifying centre vortices and then removing them from QCD ground-state fields, a deep understanding of their contributions has been developed. Removal of centre vortices from the ground-state fields results in a loss of dynamical mass generation and restoration of chiral symmetry [13,14], a loss of the string tension [15,16] and a suppression of the infrared enhancement of the Landau-gauge gluon propagator [16–18].

One can also examine the role of the centre vortices alone. Remarkably centre vortices produce both a linear static quark potential [15, 19, 20] and infrared enhancement in the Landau-gauge gluon propagator. The planar vortex density of centre-vortex degrees of freedom scales with the lattice spacing providing an well defined continuum limit [15]. These results elucidate strong connections between centre vortices and confinement.

A connection between centre vortices and instantons was identified through gauge-field smoothing [20]. An understanding of the phenomena linking these degrees of freedom was

illustrated in Ref. [21]. In addition, centre vortices have been shown to give rise to mass splitting in the low-lying hadron spectrum [13, 14, 22].

Still, the picture is not perfect. The vortex-only string tension obtained from pure Yang-Mills lattice studies has been consistently shown to be about $\sim 62\%$ of the full string tension. Moreover, upon removal of centre vortices the gluon propagator shows a remnant of infrared enhancement [18]. In the pure gauge sector, the removal of long-distance non-perturbative effects via centre-vortex removal is not perfect.

Here we turn our attention to understanding the impact of dynamical fermions on the centre-vortex structure of QCD ground-state fields. We will illustrate the differences in the microscopic structure and reveal how the change in structure affects the static quark potential and the Landau-gauge gluon propagator. We find the introduction of dynamical fermions brings the phenomenology of centre vortices much closer to a perfect encapsulation of the salient features of QCD.

## 2 Centre Vortex Identification

In identifying centre vortices one commences with a gauge fixing procedure which brings the lattice link variables as close as possible to the identity times a phase. Here, the original Monte-Carlo generated configurations are considered. They are gauge transformed directly to Maximal Centre Gauge [15, 23, 24], without preconditioning [25]. The brings the lattice link variables $U_\mu(x)$ close to the centre elements of $SU(3)$,

$$Z = \exp\left(2\pi i \frac{m}{3}\right) \mathbf{I}, \text{ with } m = -1, 0, 1. \tag{1}$$

One considers gauge transformations $\Omega$ such that,

$$\sum_{x,\mu} \left| \text{tr}\, U_\mu^\Omega(x) \right|^2 \xrightarrow{\Omega} \max, \tag{2}$$

and then projects the link variables to the centre

$$U_\mu(x) \rightarrow Z_\mu(x) \text{ where } Z_\mu(x) = \exp\left(2\pi i \frac{m_\mu(x)}{3}\right) \mathbf{I}, \tag{3}$$

and $m_\mu(x) = -1, 0, 1$.

The product of these centre-projected links around an elementary $1 \times 1$ square on the lattice reveals the centre charge associated with that plaquette. The centre-line of an extended vortex in three dimensions is identified by tracing the presence of nontrivial centre charge, $z$, through the spatial lattice

$$z = \prod_\square Z_\mu(x) = \exp\left(2\pi i \frac{m}{3}\right). \tag{4}$$

A right-handed ordering of the dimensions is selected in calculating and illustrating the centre charge. If $z = 1$, no vortex pierces the plaquette. If $z \neq 1$ a vortex with charge $z$ pierces the plaquette. We refer to the centre charge of a vortex via the value of $m = \pm 1$.

## 3 Centre Vortex Visualisation

The centre lines of extended vortices are illustrated on the dual lattice by rendering a jet piercing the plaquette producing the nontrivial centre charge. The orientation of the jet follows

**Figure 1: Illustrating nontrivial centre charge via a jet.** (left) An $m = +1$ vortex with centre charge $z = \exp(2\pi i/3)$ is rendered as a jet pointing in the $+\hat{z}$ direction. (right) An $m = -1$ vortex with centre charge $z = \exp(-2\pi i/3)$ is rendered as a jet in the $-\hat{z}$ direction.

the right-handed coordinate system. Figure 1 provides an illustration of this assignment. For example, with reference to Eq. (4), an $m = +1$ vortex in the $x$-$y$ plane is plotted in the $+\hat{z}$ direction as a blue jet. Similarly, an $m = -1$ vortex in the $x$-$y$ plane is plotted in the $-\hat{z}$ direction. As the centre charge transforms to its complex conjugate under permutation of the two dimensions describing the plaquette, the centre charge can be thought of as the directed flow of charge $z = \exp(2\pi i/3)$.

Our current focus is to understand the impact of dynamical-fermion degrees of freedom on the centre-vortex structure of a gluon field. Here we consider the PACS-CS $(2 + 1)$-flavour full-QCD ensembles [26], made available through the ILDG [27]. These $32^3 \times 64$ lattice ensembles employ a renormalisation-group improved Iwasaki gauge action with $\beta = 1.90$ and non-perturbatively $\mathcal{O}(a)$-improved Wilson quarks, with $C_{\mathrm{SW}} = 1.715$. In this section, their lightest $u$- and $d$-quark-mass ensemble identified by a pion mass of 156 MeV [26] is considered. The scale is set using the Sommer parameter with $r_0 = 0.4921$ fm providing a lattice spacing of $a = 0.0933$ fm [26].

For comparison, a matched $32^3 \times 64$ pure-gauge ensemble has been generated using the same improved Iwasaki gauge action with $\beta = 2.58$ providing a Sommer-scale spacing of $a = 0.100$ fm. This spacing facilitates comparisons with all the PACS-CS ensembles.

The centre-vortex structure of pure-gauge and dynamical-fermion ground-state vacuum fields is illustrated in Figs. 2 and 3 respectively. The vortex flow displays a rich structure. One observes the continuous flow of centre charge and the presence of monopole or anti-monopole contributions, where three jets emerge from or converge to a point. We refer to the latter as branching points in general. Upon introducing dynamical fermions, the structure becomes more complicated, both in the abundance of nontrivial centre charge and in the increased abundance of branching points.

These figures provide interactive illustrations which can be activated in Adobe Reader[1] by clicking on the image. Once activated, click and drag to rotate, Ctrl-click to translate, Shift-click or mouse wheel to zoom, and right click to access the "Views" menu. Several views have been created to facilitate and inspection of the centre-vortex structure.

Both Figs. 2 and 3 contain a percolating vortex cluster, a characteristic feature of the confining phase [28]. These illustrations are representative of the ensemble in that the vortex vacuum is typically dominated by a single large percolating cluster. This single large cluster is accompanied by several smaller loops or loop clusters. However, the most important observation is how dynamical fermions significantly increase the number of vortices observed.

For an ensemble of 200 configurations with 32 three-dimensional volume slices each, the average number of vortices composing the primary cluster in these $32^2 \times 64$ spatial slices is

---

[1]Open this pdf document in Adobe Reader 9 or later. Linux users can install Adobe acroread version 9.4.1, the last edition to have full 3D support. From the "Edit" menu, select "Preferences..." and ensure "3D & Multimedia" is enabled and "Enable double-sided rendering" is selected.

$3,277 \pm 156$ vortices in the pure gauge theory, versus $5,924 \pm 239$ vortices in full QCD. Since there are $32^2 \times 64 \times 3 = 196,608$ spatial plaquettes on these lattices, the presence of a vortex is a relatively rare occurrence.

Similarly, Figs. 4 and 5 illustrate the secondary loop structures left behind as one removes the single large percolating structure. Again, the introduction of dynamical fermions increases the complexity of the structure through a proliferation of branching points (or monopoles [29]). Figure 5 contains many views in the drop-down menu to facilitate the observation of this complexity.

## 4  Static Quark Potential

With an understanding the impact of dynamical-fermion degrees of freedom on the centre-vortex structure of ground-state vacuum fields, we turn our attention to confinement as realised in the static quark potential. The results presented here are supported by complimentary studies of the nonperturbative gluon propagator.

The static quark potential is accessed via consideration of the expectation value of Wilson loops, $\langle W(r,t) \rangle$, with spatial separation $r$ and temporal extent $t$,

$$\langle W(r,t) \rangle = \sum_{\alpha} \lambda^{\alpha}(r) \exp(-V^{\alpha}(r)t) . \tag{5}$$

The relevant static quark potential is given by the lowest $\alpha = 0$ state. We use a variational analysis of several spatially-smeared sources to isolate this state.

With knowledge of the vortex content of a configuration, contained in $Z_{\mu}(x)$ of Eq. (3), we can analyse two vortex-modified ensembles in addition to the original untouched configuration, $U_{\mu}(x)$. We refer to these as the vortex-only, $Z_{\mu}(x)$, and vortex-removed, $Z_{\mu}^{\dagger}(x)U_{\mu}(x)$, ensembles.

The static quark potential for the original untouched configurations is expected to follow a Cornell potential

$$V(r) = V_0 - \frac{\alpha}{r} + \sigma r , \tag{6}$$

composed of a Coulomb term, dominant at short distances, and a linear term, dominant at large distances. As centre vortices are anticipated to encapsulate the non-perturbative long-range physics, the vortex-only results should give rise to a linear potential [5,12,30]. On the other hand, the vortex-removed results are expected to capture the short-range behaviour. To analyse the linearity of the potential at large distances we plot a sliding local linear fit to the potential with extent $r \pm \frac{3}{2}a$.

We commence with preliminary results for the pure-gauge ensemble illustrated in Fig. 6. Qualitatively, centre vortices account for the long-distance physics. The lower plot illustrates how removal of centre vortices completely removes the long-range potential. However, the phenomenology is not perfect. The value of the string tension produced in the vortex-only analysis is once again only 60 % of the original string tension.

Upon introducing dynamical fermions with light quark masses corresponding to a pion mass of $m_{\pi} = 701$ MeV, the preliminary results shown in Fig. 7 are observed. Comparing with the pure-gauge sector of Fig. 6, we observe a screening of the original string tension with the introduction of dynamical fermions, in accord with expectations. Again, the effect of vortex removal is to remove confinement. The sliding average lies in excellent agreement with 0 at large distances.

While the centre-vortex phenomenology is similar to the pure gauge sector, this time the vortex-only string tension is in excellent agreement with the original untouched ensemble.

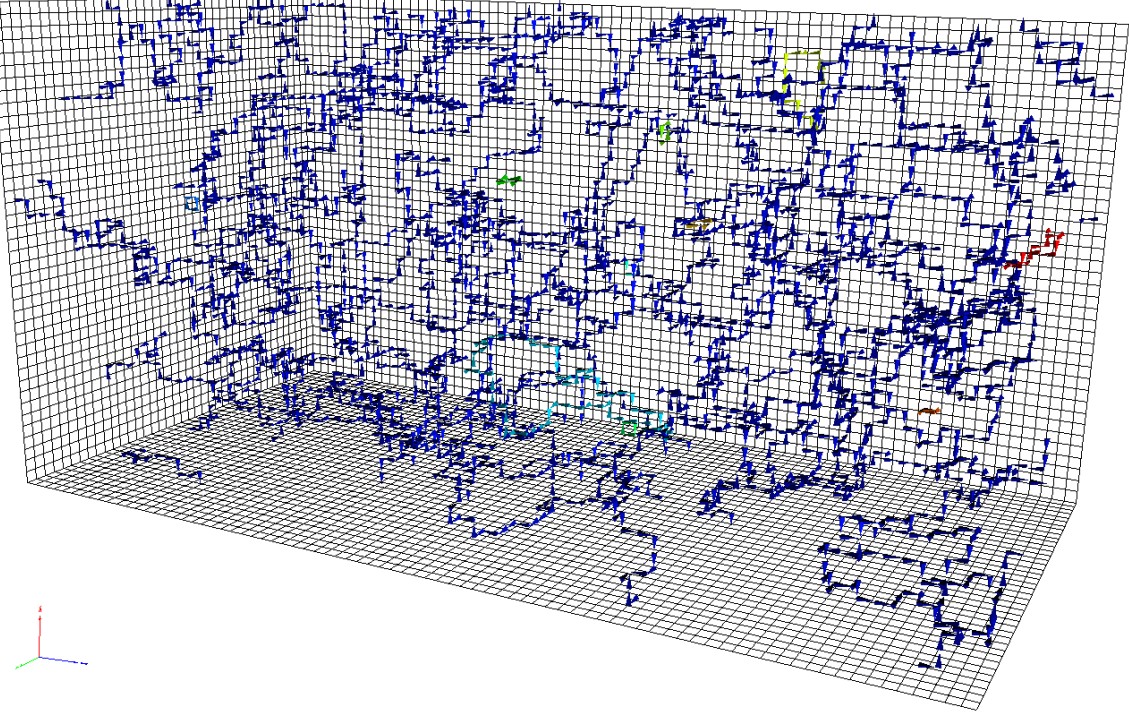

Figure 2: **The centre-vortex structure of a ground-state vacuum field configuration in pure SU(3) gauge theory.** (*Click to activate.*) The flow of +1 centre charge through a gauge field is illustrated by the jets. Blue jets are used to illustrate the single percolating vortex structure, while other colours illustrate smaller structures.

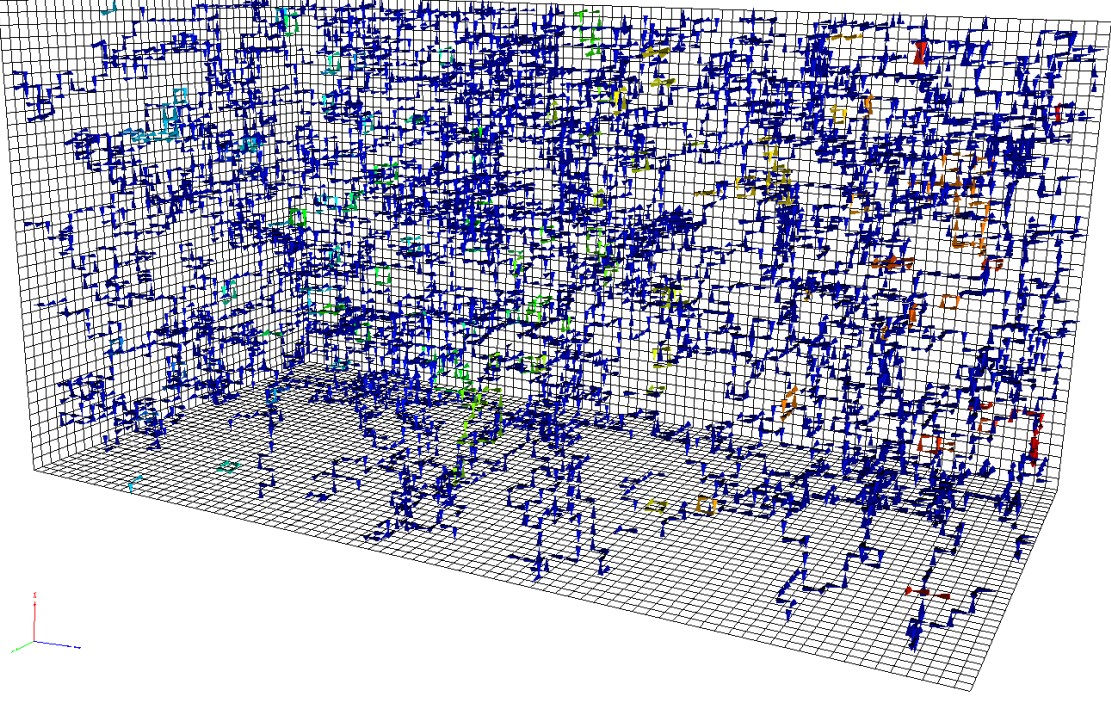

Figure 3: **The centre-vortex structure of a ground-state vacuum field configuration in dynamical 2+1 flavour QCD.** (*Click to activate.*) The flow of +1 centre charge through a gauge field is illustrated by the jets. Symbols are as described in Fig. 2.

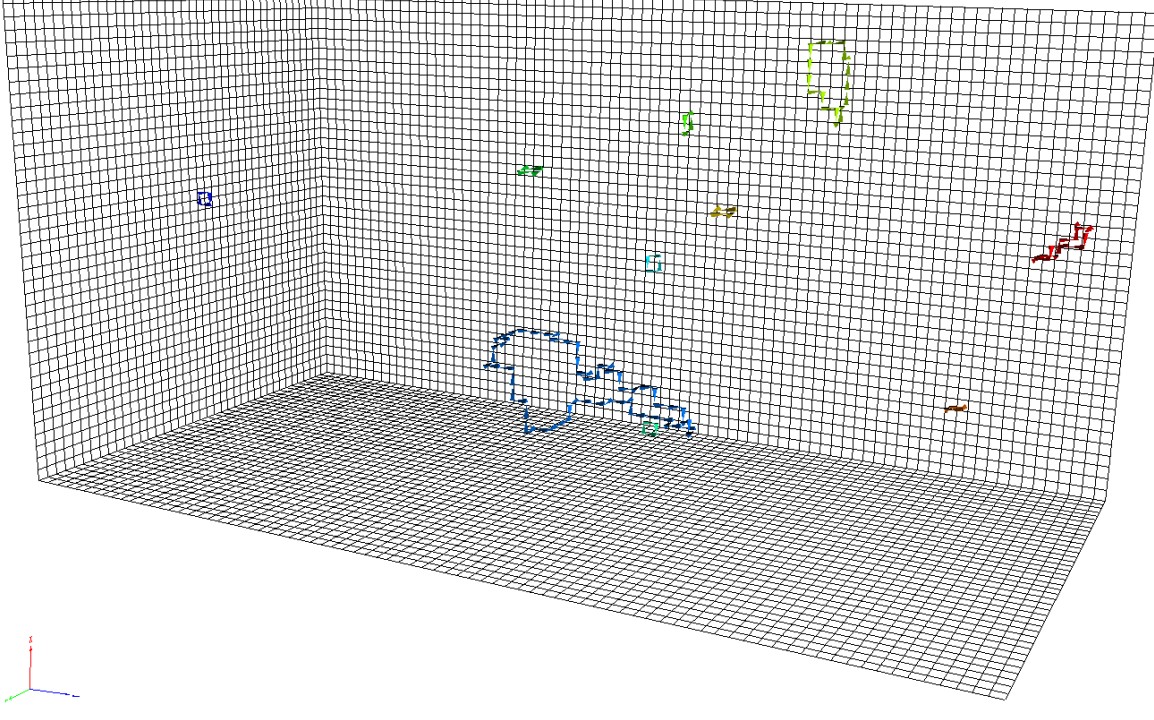

Figure 4: **The centre-vortex structure of secondary loops in a ground-state vacuum field configuration of pure SU(3) gauge theory.** (*Click to activate.*) The flow of +1 centre charge in the secondary loops – left behind as the single percolating structure is removed – is illustrated.

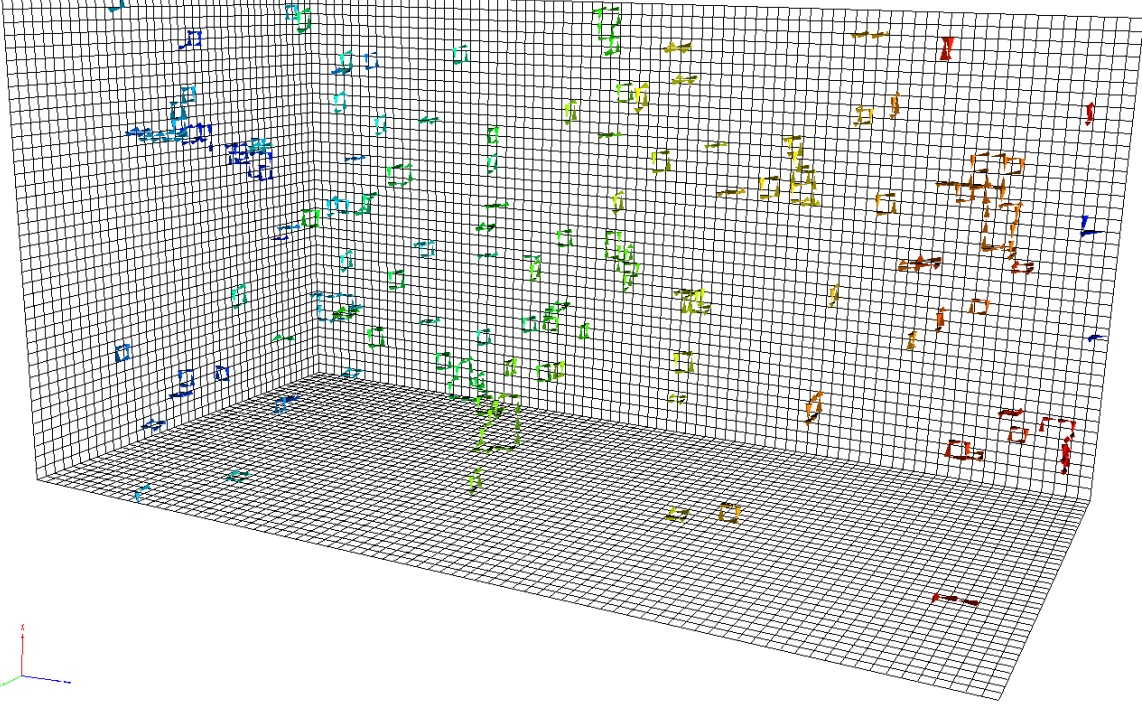

Figure 5: **The centre-vortex structure of secondary loops in a ground-state vacuum field configuration of dynamical 2+1 flavour QCD.** (*Click to activate.*) The flow of +1 centre charge in the secondary loops is illustrated.

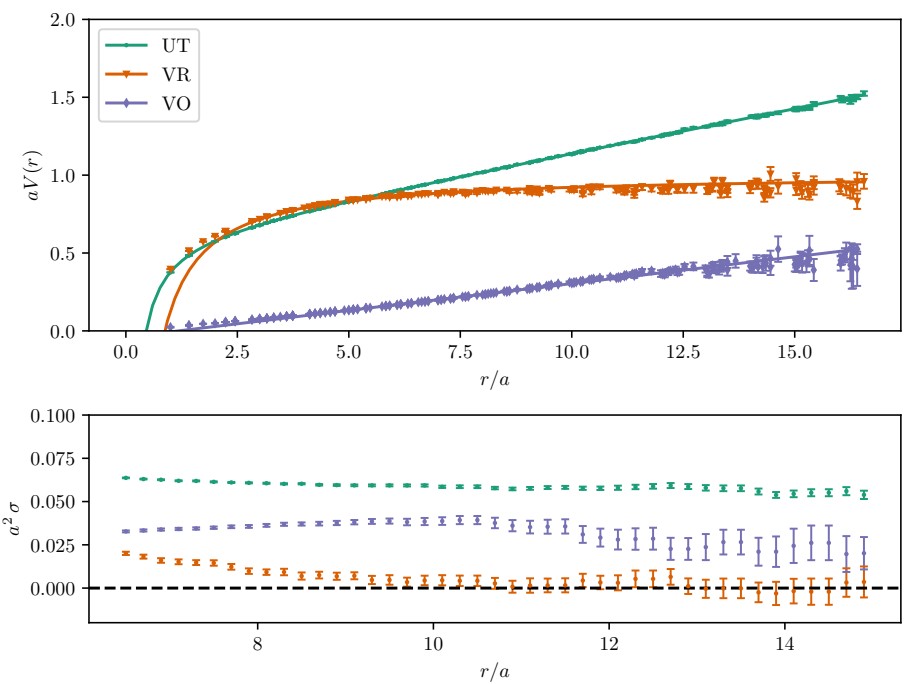

Figure 6: **The static quark potential as calculated on the original untouched and vortex-modified pure-gauge ensembles.** The lower plot shows the local slope of the potentials at position $r$ obtained from a linear fit with extent $r \pm \frac{3}{2}a$.

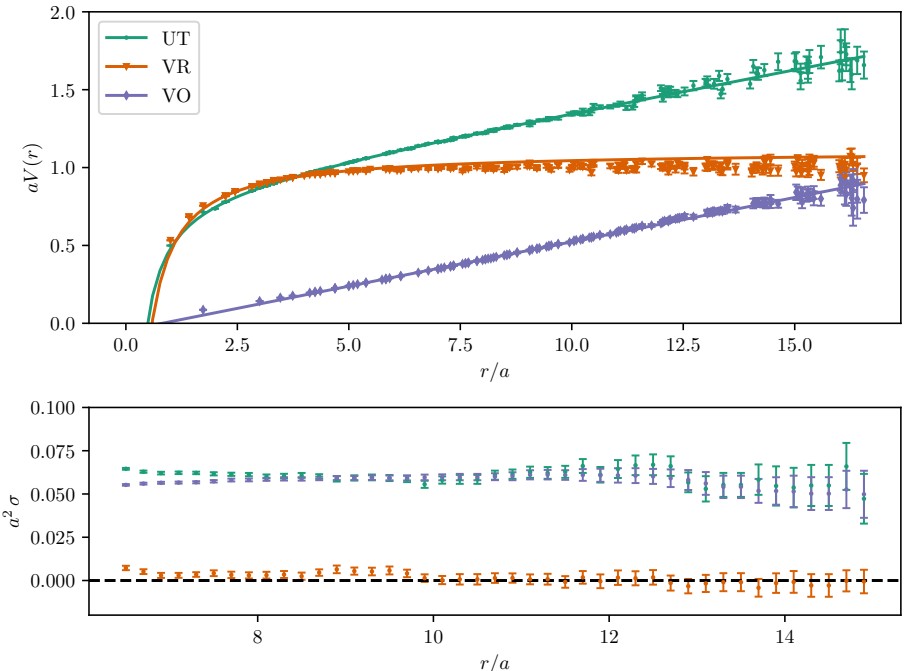

Figure 7: **The static quark potential as calculated on the vortex-modified dynamical-fermion ensembles, corresponding to a pion mass of** 701 **MeV.** Details are as in Fig. 6.

This is illustrated in Fig. 7, in the lower plot where the local slopes of the untouched and vortex-only ensembles agree at large distances. This new agreement arises from significant modifications in the centre-vortex structure of ground state fields induced by dynamical fermions, even at relatively large quark masses.

## 5 Conclusion

In summary, centre-vortex structure is complex. Each ground-state configuration is dominated by a long-distance percolating centre-vortex structure. In $SU(3)$ gauge field theory, a proliferation of branching points is observed, with further enhancement as light dynamical fermion degrees of freedom are introduced in simulating QCD. There is an approximate doubling in the number of nontrivial centre charges in the percolating vortex structure as one goes from the pure-gauge theory to full QCD. An enhancement in the number of small vortex paths is also observed upon introducing dynamical fermions. Increased complexity in the vortex paths is also observed as the number of monopole-antimonopole pairs is significantly increased with the introduction of dynamical fermions. In short, dynamical-fermion degrees of freedom radically alter the centre-vortex structure of the ground-state vacuum fields.

With regard to the static quark potential and confinement, we find that centre vortices now quantitatively capture the string tension in full QCD, unlike the pure-gauge sector. This represents a significant advance in centre-vortex phenomenology. Moreover, vortex removal continues to eliminate the long distance potential. These encouraging results are also reflected in more recent studies of the gluon propagator in full QCD. In summary, the results presented here show a significant advance in the ability of centre vortices to capture the salient nonperturbative features of QCD.

## Acknowledgements

We thank the PACS-CS Collaboration for making their 2 +1 flavour configurations available via the International Lattice Data Grid (ILDG).

**Funding information**   This research was undertaken with the assistance of resources from the National Computational Infrastructure (NCI), provided through the National Computational Merit Allocation Scheme and supported by the Australian Government through Grant No. LE190100021 via the University of Adelaide Partner Share. This research is supported by Australian Research Council through Grants No. DP190102215 and DP210103706. WK is supported by the Pawsey Supercomputing Centre through the Pawsey Centre for Extreme Scale Readiness (PaCER) program.

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
