# Peer review of "Centre Vortex Structure of QCD-Vacuum Fields and Confinement"

_SciPost Physics Proceedings, doi:SciPost Phys. Proc. 6, 004 (2022)_

## Round 1 · Referee Report · Anonymous · 2022-3-17

Strengths

Very nice and clear presentation of results
Very elaborated use of visualization
New results obtained for the center dominance phenomena

Weaknesses

Minor misprints

Report

The paper is devoted to the fundamental still unsolved problems of the non-perturbative QCD - the problem of confinement of quarks. There are various proposals for the confinement scenarios. The authors pursue the idea first formulated by t'Hooft that confinement is due to the center vortex configurations of the gauge field. Their subject is the impact of dynamical fermions on the properties of the center vortices.
In the Introduction the authors review the results of the numerous earlier studies of the center vortices properties and their role in the nonperturbative QCD. Section II is devoted to the definitions of the center gauge and center vortices. In section III the special features (lattice size, pion mass, lattice spacing, etc.) of the full QCD lattice configurations used in the work are explained. This section is devoted to a comparison of the center vortices in the full lattice QCD and in the lattice gluodynamics via visualization. It is observed that the dynamical fermions significantly increase the number of vortices. In section IV the static potential is discussed. The original static potential is compared with the static potential obtained from the center projected field and from the nonabelian field with removed vortices. The main observation is that the static potential obtained from the center projected field in full QCD has the same string tension as the original static potential. This is quite different from the quenched theory where the projected string tension is only 60% of the original one.
In my opinion, the paper presents new interesting results of the confinement problem studies. It definitely deserves to be published.

Requested changes

It is necessary to correct few misprints
1) a lattice spacing of a = 0.933 fm [26].
2) linking these these degrees of freedom

---

## Editorial Decision

published